# Optimization of *Typha* Fibre Extraction and Properties for Bio-Composite Applications Using Desirability Function Analysis

**DOI:** 10.3390/polym14091685

**Published:** 2022-04-21

**Authors:** Mahmudul Hasan, Mashiur Rahman, Ying Chen, Nazim Cicek

**Affiliations:** 1Department of Biosystems Engineering, University of Manitoba, Winnipeg, MB R3T 2M8, Canada; hasanm37@myumanitoba.ca; 2E1-349 EITC, Department of Biosystems Engineering, University of Manitoba, Winnipeg, MB R3T 5V6, Canada; ying.chen@umanitoba.ca (Y.C.); nazim.cicek@umanitoba.ca (N.C.)

**Keywords:** *Typha* fibre, waste biomass fibre, optimum extraction bath parameters, desirability function analysis (DFA), bio-composites

## Abstract

The effect of extraction time, temperature, and alkali concentration on the physical and mechanical properties of cattail (*Typha latifolia* L.) fibres were investigated using five levels of time (4, 6, 8, 10, and 12 h), four levels of temperature (70, 80, 90, and 95 °C), and three levels of NaOH concentration (4, 7, 10%, *w*/*v*) in a 3 × 4 × 5 factorial experimental design. The extraction parameters were optimized for bio-composite application using a desirability function analysis (DFA), which determined that the optimum extraction time, temperature and NaOH concentration were 10 h, 90 °C, and 7%, respectively. A sensitivity analysis for optimal treatment conditions confirmed that the higher overall desirability does not necessarily mean a better solution. However, the analysis showed that the majority of optimum settings for time, temperature, and concentration of NaOH found in the sensitivity analysis matched with the optimum conditions determined by DFA, which confirmed the validity of the optimum treatment conditions.

## 1. Introduction

In recent decades, the use of composite materials has increased rapidly, and the desire for lightweight yet stronger materials has accelerated the development of fibre-reinforced polymer composites [1,2] over traditional wood materials [3]. Synthetic fibres, i.e., glass, carbon, and aramid, are the most widely used to reinforce polymer composites due to their excellent mechanical properties [1]. However, they are non-biodegradable, and their manufacturing processes contribute an enormous amount of chemical waste and carbon emissions [4,5]. These negative environmental impacts of synthetic fibres have drawn attention to the development of environmentally friendly natural biomass fibre reinforced bio-composites [6].

However, natural biomass fibres such as flax, hemp, jute, sisal, kenaf, coir, kapok, and banana have very low yield (5–15%), poor water resistance, low durability, and poor fibre-matrix interfacial bonding, which cause a diminution in the final properties of the composites and ultimately inhibit their industrial applications [7,8,9]. These limitations, particularly of supply constrain of existing natural biomass fibres, have accelerated the research activities to examine the feasibility of naturally available waste biomass fibres such as cattail fibres for industrial applications [9,10].

*Typha* spp., commonly known as cattail, is the dominant wetland plant in North America [11]. Mainly three species of *Typha* are found in this region: *Typha latifolia* L., *Typha angustifolia* L., and the hybrid of these two *Typha × glauca*. *Typha* spp. possess some unique characteristics. They populate rapidly, are hydrophilic, biodegradable, and abundant [12]. Studies have revealed that *Typha latifolia* L. fibres blended with wheat straw exhibited acceptable mechanical properties for composite panels [13]. Moreover, the cellulose in the *Typha latifolia* L. can be extracted to produce cellulose nanofibres which have potential applications in packaging, fibre reinforced composites, and electronic displays [14]. All these findings show that the fibres obtained from the *Typha latifolia* L. have great potential in fibre-reinforced bio-composites and in other industrial applications.

So far, different retting methods have been applied to extract fibres from the leaves of *Typha latifolia* L., such as water retting, alkali retting, enzyme retting, acid retting, and alcohol retting [9,15]. Out of these methods, alkali treatment, particularly treated with NaOH, was found to be the only method that is most effective and economical and possesses the least environmental impacts [9,15,16].

The studies have revealed that the extracted fibres obtained from the *Typha latifolia* L. possess similar characteristics to commonly used cellulosic fibres such as hemp, jute, and flax in terms of mechanical properties, fibre diameter, moisture regain, and thermal properties [9,15]. All these properties are prerequisites in order to be used as reinforcements in fibre-reinforced composites. Moreover, the fibre yield (%) of broad-leaf cattail was found to be significantly higher than other common natural fibres [9]. Therefore, fibres obtained from the broad-leaf cattails can be an ideal source to produce ‘novel’ fibre-reinforced bio-composites which may be used in automotive industries, packaging industries, construction materials, furniture and home décor materials.

As a result, it is necessary to know how the extraction parameters (extraction time, temperature, and alkali concentration) affect the fibres properties and fibre yield (%) in order to determine the optimum processing parameters that would produce fibres of desired characteristics for a specific composite application, which is the primary objective of this study.

In the present research, a full factorial analysis of different processing times, temperatures, and concentration of NaOH on fibre yield (%) and fibre properties (diameter, strength, modulus, extension at the break, moisture regain) was performed to validate the usability of cattail fibres as a composite reinforcement. Finally, the fibre extraction method was optimized using Desirability Functions Analysis (DFA) with a sensitivity test.

## 2. Materials and Methods

### 2.1. Materials

The plant material of *Typha latifolia L*. was harvested from the Assiniboine Forest area, in Winnipeg, MB (Canada), in late October, 2017 by using a knife.

### 2.2. Methods

In this study, 3 levels of concentration (4, 7, 10%, *w*/*v*) of NaOH solution expressed as ‘C’ were used with 4 levels of temperature (70, 80, 90 and 95 °C) expressed as ‘H’ and 5 levels of time (4, 6, 8, 10, and 12 h) expressed as ‘t’ in a 3 × 4 × 5 (total treatment 60) factorial experimental setup. The number of replications per treatment and the total number of prepared samples are illustrated in the following Table 1.

### 2.3. Fibre Extraction

The leaves of *Typha Latifolia* L. were cut to 4 cm length and then dried in the oven at 105 °C. The oven was equipped with a ventilator and a thermostat, and the dehydration by heating was carried out for 8 h until a constant dry weight was achieved. Desiccation was considered complete when the difference in weights in a sample a was obtained less than 0.05% when weighed two successive times at 15 min interval. The weighing was carried out using an electronic balance.

For each treatment, 4 samples (0.40 and 0.42 g) were placed in 4 different Erlenmeyer flasks. In each flask, 200 mL of either 4/7/10% (*w*/*v*) concentrated sodium hydroxide solution was added. The extraction treatment was carried out for a set time in temperature-controlled water bath. After the treatment, fibres were thoroughly washed with distilled cold water and neutralized with acetic acid to remove residual alkali and dissolved substances.

### 2.4. Yield Measurement

The gravimetric method was used to calculate fibre yield (%) using the formula expressed in Equation (1).
(1)$$\mathrm{Yield}\;(\%)=\frac{M_{a}}{M_{b}}\times 100$$
where, $M_{a\;}$ = oven-dried mass of chemically extracted fibres, and $M_{b\;}$ = oven-dried mass of *Typha* leaves before chemical treatment.

### 2.5. Diameter Measurement

Ten single fibres from each treatment (a total of 600 fibres) were randomly separated for diameter measurement using Bioquant Analyzer (Bioquant Life Sciences, Nashville, TN, USA). The diameters were measured in five different places across the length, and the average diameter was reported for each fibre. This was necessary due to the variations in diameter across the length of the single fibre, which is a common phenomenon in natural fibres [17].

### 2.6. Mechanical Properties Measurement

The mechanical properties, i.e., tensile strength, modulus of elasticity, and elongation at break (%), were measured using the Instron Tensile Tester (Model 5965, Bluehill^®^ 2 (Norwood, MA, USA) following the ASTM D3822 method. Before the measurements, all the fibre samples were conditioned at 21 °C temperature and 50% relative humidity for 48 h, as shown in Figure 1. After the conditioning was performed, the frame holding the single fibre was placed between the jaws of the machine in a way (as shown in Figure 2) that the inside length of the frame acted as a gauge length. Then the vertical stands of the frame were cut with a scissor, and the length of the fibre inside the frame acted as a gauge length which is approximately 25 mm.

However, the fibres acquired some crimp during the fibre extraction and washing, and it was necessary to remove the crimp to determine the actual length of the fibre attached inside the frame. For this reason, ‘Pretest’ and ‘Auto-balance’ functions in the ‘Instron Bluehill 2’ software, Norwood, MA, USA, were used. The ‘Pretest’ function allows the machine to extend the fibre, but no data are reported until a small amount of load is experienced by the load cell (which was chosen as 0.3 N for this experiment). When the load cell experiences the specified amount of load, the extension up to this point is considered due to the crimp, and the ‘Auto-balance’ function adds this length with the initial length of the fibre, and then the original test begins. All the 600 fibre samples were tested using this procedure. For this experiment, the crosshead moved at a speed of 20 mm/min, and a 5 N load cell was used to conduct the tensile strength tests. All the results (tensile strength, modulus of elasticity, and elongation at break) and graphs for each fibre were obtained directly from the ‘Instron Bluehill 2’ software.

### 2.7. Moisture Regain (%) Measurement

For measuring the moisture regain, 3 fibre bundles from each treatment, a total of 180 samples from 60 treatments were randomly taken according to ASTM D 1776. The oven-dry constant weight of the samples was measured, and then the fibres were conditioned in a humidity chamber (21 ± 1 °C and 65 ± 3%) in the Agriculture Canada laboratory located at the University of Manitoba. The moisture regains (%) were measured as a percentage of the ratio of the weight of water absorbed by the sample to the oven-dry weight of the sample as expressed in Equation (2).
(2)$$\mathrm{Moisture}\;\mathrm{Regain}\;(\%)=\frac{M_{w\;}-M_{o}}{M_{o}}\times 100$$
where, $M_{w\;}$ = weight of the samples after conditioning, and $M_{o\;}$= Oven dry weight of the samples.

### 2.8. Statistical Analysis

The experimental design was a full factorial design that included all levels of each factor (time, temperature, and concentration of NaOH) which were present in combinations of all levels of other factors [18]. In this analysis, *t*, *H*, and *C* were used as input variables, whereas fibre yield (%), diameter, moisture regain (%), and mechanical properties (tensile strength, extension at break, and modulus of elasticity) were considered as output variables. All statistical analysis was performed using the SAS^®^ University Edition software (2019 Edition) created by SAS Institute, Cary, NC, USA.

The model used for analysis was a fixed model as all the levels of time, temperature, and concentrations of NaOH were predetermined. The model that has been used to perform the three-way factorial ANOVA is shown in Equation (3):*Yijkl = µ + Ti + tj + Ck + tTij + tCjk + TCik + tTCijk + eijkl*(3)
Here,
*Yijkl* = l’th response treated with i’th temperature, j’th treatment time, and k’th concentration of NaOH.*µ* = population mean*Ti* = effect of i’th treatment temperature on l’th response.*tj* = effect of j’th treatment time on l’th response.*Ck* = effect of k’th concentration of NaOH on l’th response.*tTij* = effects of interaction of time and temperature on l’th response.*tCjk* = effects of interaction of time and concentration on l’th response.*TCik* = effects of interaction of temperature and concentration on l’th response.*tTCijk* = effects of interaction of time, temperature, and concentration on l’th response.*eijkl* = error variations.Range-*J* = 70 to 95 °C*i* = 4 to 12 h.*k* = 4 to 7%*l* = 1 to 4 for yield (%)Number of replications = 1 to 10 for diameter, tensile strength, modulus of elasticity, elongation at break (%)Number of replications = 1 to 3 for moisture regain (%)

After performing ANOVA, the non-significant effects were removed to simplify the model.

## 3. Results and Discussion

### 3.1. Background

Before conducting the three-way factorial ANOVA of fibre yield (%), and fibre properties, the distribution of each response variable was examined. The skewness and kurtosis values of the data of each response variable and the distribution used for analysis are summarized in Table 2.

After determining the appropriate distribution, a three-way factorial ANOVA of fibre yield (%) and fibre properties was performed, as shown in Table 3. For pairwise comparison of the treatment means, Tukey–Kramer post hoc procedure was followed at α = 0.05.

The main effects of time, temperature, and concentration are significant for all properties except for modulus with concentration (C), while the effects of interaction between t × C were significant (*p* < 0.05) for modulus and between H × C for all properties measured (Table 3). The interaction of H × t × C is not significant (Table 3).

### 3.2. Physical Properties of Fibre

The effects of the interaction of temperature and concentration for these physical properties of fibres are combined and illustrated in Figure 3. From this figure, it can be observed that at the 4% NaOH concentration level, both fibre yield (%) and diameter progressively decreased with an increase in temperature. This decrease was significant (*p* < 0.05) from 70° to 80 °C. However, from 80 °C, the reduction in yield (%) and fibre diameter was not significant (*p* > 0.05) up to 95 °C. At the 7% NaOH concentration level, the decrease in fibre yield (%) was found significant between 70° and 95 °C (*p* = 0.044). For fiber diameter at the 7% concentration level, the diameter progressively decreased between 70° and 90 °C (*p* < 0.001) and 80° and 90 °C (*p* < 0.001). However, no significant change (*p* > 0.05) in diameter was observed from 90° to 95 °C, and, finally, at a 10% concentration level, no significant change (*p* > 0.05) in fibre yield (%) and fibre diameter was observed for all temperatures (70°, 80°, 90°, and 95 °C).

*Typha* fibre has a composite structure where cellulosic fibre bundles of different sizes and numbers are linked together by gummy and waxy substances composed of pectin (mostly), lignin, hemicellulose, wax, and fat materials [15,22]. Sana et al. [14] observed that the alkali treatment removed the wax and fatty substances from the surface of cattail fibres layer by layer, decreasing the fibre yield (%) and diameter. As a result, when considering the effects of the interaction of temperature and concentration (Figure 3), it was found that at both 4% and 7% NaOH concentration levels, the decrease in both fibre yield (%) and diameter was significant with increase in temperature. However, at a 10% concentration level, the change in yield (%) and diameter was not significant from 70 to 95 °C (*p* > 0.05). This could be due to the fact that, at high alkali concentrations (10%), the removal of impurities was already higher at 70 °C, and the increase in temperature (up to 95 °C) did not have any significant effect.

For moisture regain (%) of cattail fibres, at a 4% concentration level, it decreased significantly (*p* = 0.001) with an increase in temperature from 70° to 95 °C. Similarly, a significant decrease in moisture regain (%) was observed from 70° to 80° (*p* = 0.001) at 7% concentration level, and from 70° to 90 °C (*p* = 0.002) at 10% concentration level. Therefore, high temperature, longer treatment duration, and high concentration of NaOH decreased the moisture regain (%) of cattail fibres. These findings match with previous results, which revealed that alkali treatment of natural cellulosic fibres lowers the moisture absorption due to the removal of lignin and hemicellulose components from the composite structure [20,21,22,23,24]. With increased temperature, time, and concentration of NaOH, the removal of impurities (lignin, pectin, hemicellulose) was higher, which may have contributed to the decrease in moisture regain (%). However, the small but insignificant (*p* = 0.083) increase in moisture regain (%) from 90° to 95 °C at 10% NaOH may arise due to the inherent variability of naturally sourced materials.

### 3.3. Mechanical Properties of Fiber

The Stress Vs Strain diagram of thirty controlled cattail fibres is shown in Figure 4. It appears that the stiff and strong fibres seem to have low elongation. As discussed earlier, alkali treatment removes impurities and improves the molecular orientation of the fibres and therefore, increasing the crystallinity of the fibres. As a result, fibres become stiffer with a hard crystalline structure, resulting in a decrease in the elongation at break (%) [25,26]. Moreover, it can be noted by observing the relative arrangement of curves that there is a great variation of tensile properties among the fibres with high values of the coefficients of variation. Unlike synthetic fibres, this is a common phenomenon in the case of natural cellulosic fibres (Coefficient of Variation: ~30–45%) [27].

The effects of interactions on the mechanical properties of cattail fibres are combined and illustrated in Figure 5. At a 4% concentration level of NaOH, the tensile strength of fibres gradually increased with an increase in temperature, which became highly significant (*p* < 0.001) between 70° and 95 °C (Figure 5). However, at 7% and 10% concentration levels, the increase in temperature did not have any significant effects (*p* > 0.05) (Figure 5). For elongation at break (%), at a 4% concentration level, the change in elongation of the fibres was not significant. However, the elongation of the fibres decreased significantly from 80 °C to 95 °C (*p* = 0.004) at 7% concentration level, and from 70 °C to 95 °C (*p* < 0.001) at 10% concentration level.

Previous studies have revealed that the tensile properties (strength, fineness) of cellulosic fibres increase with the concentration of NaOH and temperature due to the removal of impurities and the rearrangement of cellulosic chains [15,22,28]. This may be the reason for the highly significant increase in tensile strength from 70° to 95 °C at a 4% concentration level. However, at both 7% and 10% concentration levels, no significant differences (*p* > 0.05) in tensile strength were found from 70 to 95 °C. This may be due to the fact that a high concentration of NaOH (7 and 10%) caused greater removal of impurities at 70 °C, and the increase in temperature up to 95 °C did not cause any significant effects. Moreover, higher removal of impurities improved the molecular orientation of the fibres and, therefore, increased the crystallinity of the alkali-treated fibres. As a result, fibres become stiffer with a hard crystalline structure [26]. This resulted in the decrease in elongation at break (%) at high concentrations of NaOH and temperature.

From the interaction between time and concentration for modulus of elasticity (Figure 5), it can be seen that at both 4% and 7% concentration levels, the change in modulus of the fibres was not significant (*p* > 0.05) with the increase in treatment time. However, at 10% concentration levels, with an increase in treatment time from 4 h to 12 h, the modulus of elasticity of the fibres increased significantly (*p* = 0.002). Furthermore, the interaction between temperature and concentration showed that at a 4% concentration level, the increase in temperature from 70 °C up to 95° did not have any significant effect on the modulus of fibres. However, the modulus of fibres increased significantly from 80° to 95 °C (*p* < 0.001) at 7% concentration level, and from 90° to 95 °C (*p* = 0.002) at 10% concentration level.

Overall, high concentration, high temperature, and longer treatment time improved the modulus of elasticity of cattail fibres. As discussed earlier, alkali treatment of cellulosic fibres improves the molecular orientation of cellulose and removes the impurities and, therefore, increases the crystallinity of the fibres [29]. At a high temperature, high concentration, and longer treatment time, this process becomes faster, yielding thinner fibres with an improved modulus of elasticity.

### 3.4. Potential Composite Applications of Cattail Fibres

A comparison of properties of cattail fibres with other commonly used natural fibres, i.e., flax, hemp, sisal, and coir used for composite applications, are shown in Table 4 [6,30,31]. From the table, it is evident that cattail fibres showed lower tensile strength and modulus of elasticity compared to flax and hemp fibres. However, it showed similar tensile strength as coir fibres, and the modulus of elasticity was found to be higher than coir fibres and almost similar to sisal fibres. Therefore, similar to sisal and coir fibre composites, the cattail fibre composites have potential applications in the automotive and packaging industry [29].

The major advantages of cattail fibres are an abundance of supply without any cost of growing and significantly higher fibre yield (%). Furthermore, cattail does not require any water during plant growth, reducing greenhouse emissions. However, the moisture regains of cattail fibre is high, which is comparable with flax and hemp. Therefore, for composite applications, further surface treatment is required to enhance adhesion with the hydrophobic resins [32].

#### 3.4.1. Optimizing Fiber extraction Process with Desirability Function Analysis

Desirability function analysis (DFA), popularized by Derringer and Suich [33], is one of the most widely used methods for process optimization having multiple responses. The desirability function transforms each estimated response variable $“{\hat{Y}}_{i}”$ to a desirability value $d_{i},$ where $0\;\leq \;d_{i}\;\leq 1$. The individual desirability is then combined using the geometric mean.

Here, two different types of desirability functions $“d_{i}”$ were used, i.e., desirability function to maximize, and desirability function to minimize. For maximizing a property $“Y_{i}”$, the desirability function $\left(d_{i}\right)$ was calculated using the following Equations (4)–(6):(4)$$d_{i}=0\;\mathrm{if}\;{\hat{Y}}_{i}<Y_{min}$$
(5)$$d_{i}={\left[\frac{{\hat{Y}}_{i}-Y_{min}}{C-Y_{min}}\right]}^{s}\mathrm{if}\;Y_{min}\leq {\hat{Y}}_{i}<C$$
(6)$$d_{i}=1\;\mathrm{if}\;{\hat{Y}}_{i}\;\geq \;C$$
where *C* is the upper criteria value or the requirement, *Y_min_* is the lower tolerance value, and s represents weight. When ${\hat{Y}}_{i}$ equals or exceeds the upper criteria value, which is the requirement, the desirability function equals 1. When ${\hat{Y}}_{i}$ is less than the lower tolerance value, which is unacceptable, the desirability function equals to 0.

For minimizing a property $“Y_{i}”$, the desirability function $\left(d_{i}\right)$ was calculated using the following Equations (7)–(9):(7)$$d_{i}=1\;\mathrm{if}\;{\hat{Y}}_{i}\;\leq C$$
(8)$$d_{i}={\left[\frac{{\hat{Y}}_{i}-Y_{max}}{C-Y_{max}}\right]}^{t}\mathrm{if}\;C<{\hat{Y}}_{i}\;\leq \;Y_{max}$$
(9)$$d_{i}=0\;\mathrm{if}\;{\hat{Y}}_{i}>Y_{max}\;$$
where *C* is the lower criteria value or the requirement, *Y_max_* is the upper tolerance value, and *t* represents weight. When *Y_i_* is equal to or less than the lower criteria value, which is the requirement, the desirability function equals 1. When *Y_i_* exceeds the upper tolerance value, which is unacceptable, the desirability function equals 0. Therefore, before calculating the individual desirability function, the objective of each property, the criteria value, the tolerance value, and the weights were fixed. The values of *s* and *t* are specified by the user, and Derringer and Suich [33] suggested that a large value of weights would be specified if the property (*Y_i_*) is very desirable.
(10)$$d_{G}=\sqrt[{w}]{d_{1}{}^{w_{1}}\times d_{2}{}^{w_{2}}\times d_{3}{}^{w_{3}}\times \ldots \times d_{n}{}^{w_{n}}}\;$$
where i ∈[1…n], $d_{i}$ is the individual desirability of the property $Y_{i}$, $w_{i}$ is the relative importance of the property $“Y_{i}”$ in the composite desirability $\left(d_{G}\right)$ shown in Equation (10), w is the sum of the individual importance $\left(w_{i}\right)$, and n is the number of properties. Therefore, this single value of composite desirability represents the overall assessment of the desirability of the combined response levels [33]. The value of $d_{G}$ falls within the range of 0 to 1, and the value of $d_{G}$ increases as the combination of properties become more favourable. Moreover, if any $d_{i}=0$, meaning one of the response variables is unacceptable; the value of $d_{G}$ becomes 0, which implies that the overall product is unacceptable.

For bio-composite applications, the objectives of tensile strength and modulus of elasticity of the fibres are to maximize, and the objectives of fibre diameter and moisture regain (%) are to minimize [34,35]. The optimum values of the physical and the mechanical properties of extracted cattail fibres obtained from this study are shown in Table 5.

The cattail fibres showed a relatively low elongation at break (%), and the estimated means ranged between 1.16–2% (Table 5). Due to the small range, this property was not included in the list of optimized properties. The objectives of fibre properties, the criteria/target values, the tolerance values, weight of individual property, the importance of co-efficient of each property compared to others, and the reference values are listed in Table 6. The reference values were taken from the properties of commonly used natural fibres used in composite applications [29,30].

For composite applications, it is desirable for tensile strength and modulus of elasticity to attain the optimum values, and, therefore, higher weights are given to these properties. Furthermore, higher importance coefficients are given to fibre diameter, tensile strength, and modulus as these are primary characteristics affecting the overall performance of composites [29].

The desirability index of individual property ($d_{i})$ was calculated by selecting Equations (3)–(8). The composite desirability $\left(d_{G}\right)$ for each treatment by combining all the individual desirabilities were calculated using Equation (9). The highest composite desirability value (0.796, data not shown here) was obtained from the fibres treated with a 7% (*w*/*v*) concentration of NaOH at 90 °C treatment temperature for 10 h treatment duration. Therefore, the cattail fibres obtained from this treatment would be most suitable for bio-composite applications, and the optimum values or the estimated treatment means of yield (%), diameter (µm), tensile strength (MPa), modulus of elasticity (GPa), and moisture regain (%) for this treatment are 24.08, 79.83, 165.56, 14.01, and 8.73, respectively.

#### 3.4.2. Sensitivity Test of DFA for Composite Applications

At present, there is no standard that can be used to select or determine the parameters of desirability functions. Therefore, the selection of these parameters is susceptible to bias or arbitrary choices [37]. In general, these parameters are determined based on applications and manufacturing costs [33,38]. However, from a design and quality perspective, the selection of parameters should have some statistical basis so that the optimization results can be analyzed further [37]. This analysis should examine the robustness of overall desirability to changes in these parameters.

In this study, a sensitivity analysis of desirability functions was conducted by following the approach proposed by Aksezer [39]. The upper and lower edges of each parameter were assigned as shown in Table 7. Each parameter was designated by a letter A to O. For sensitivity analysis, 15 factors which are the importance coefficients, weights, and ranges of responses, were analyzed. The selected factors were investigated by a 2-level design. The Plackett–Burman design was selected for this sensitivity analysis as only the main effects are required to be examined to draw relevant conclusions [39]. Furthermore, this design is very useful for detecting the main effects with a smaller number of experiments as this design can examine up to the N-1 number of factors for N experiments.

For this analysis, the Plackett–Burman design with 20 runs was selected. The experimental setup and the resulting values for overall desirability are shown in Table 8. All the calculations were conducted using the JMP^®^ 14.2. software. The resulting ANOVA is given in Table 9. The overall desirability model is found to be significant with a *p* < 0.05 and an acceptable adjusted R-square value of 0.96. The prediction equation on the overall desirability is found as given in Equation (10).
(11)Overall desirability=0.559−0.076(A)−0.034(B)−0.029(C)−0.017(D)−0.025(E)−0.016(F)+0.012(G)+0.015(H)+0.034(I)−0.025(J)+0.193(K)+0.042(L)−0.004(M)+0.001(N)+0.103(O)

After analyzing the parameter estimates, it was found that the range in (%) yield (Designation: *K*) is the most sensitive parameter, followed by the weight of (%) yield (Designation: *A*). The positive sign indicates that any increase in *K* will increase the overall desirability rapidly, and the negative sign indicates that a decrease in A will increase the overall desirability. Therefore, selecting a moderate range for yield (%) between 20–34% and keeping the weight for moisture regain (%) to 1 while determining the optimum parameters for composites was a prudent decision. Furthermore, the range of tensile strength (Designation: *M*) and modulus of elasticity (Designation: *N*) proved to be the least sensitive. The proposed procedure allowed the sensitivity of the important characteristic parameters of the desirability function and their impact on the optimal solution to be analyzed. However, it should be noted that optimum levels of input variables (time, temperature, and concentration of NaOH) obtained from different design points ended up with equal settings with different overall desirability levels, which proved that the higher overall desirability does not necessarily mean a better solution.

## 4. Conclusions

The estimated treatment means of tensile strength and modulus of elasticity obtained for cattail fibres from the 3-way factorial ANOVA was found similar to commonly used natural fibres for composite applications.

The fibre extraction parameters were optimized using DFA, which calculated the optimum extraction parameters for automobile applications as: 7% (*w*/*v*) concentration of NaOH, 10 h treatment duration, and 90 °C treatment temperature. The cattail fibres obtained from this treatment would be most suitable for bio-composite applications.

A sensitivity analysis of desirability functions was performed. This test allowed the effects of the range of individual properties, the weights of the properties, and the importance coefficient of each property in the composite desirability to be analyzed. The analysis showed that the optimum treatment conditions could be changed by changing the range, weight and importance coefficient of individual properties. Therefore, it should be noted that higher composite desirability does not necessarily mean better treatment. However, the analysis showed that the majority of optimum settings for time, temperature, and concentration of NaOH found in the sensitivity analysis matched the optimum conditions determined for bio-composite applications, which confirmed the validity of the optimum treatment conditions.

## Figures and Tables

**Figure 1 polymers-14-01685-f001:**
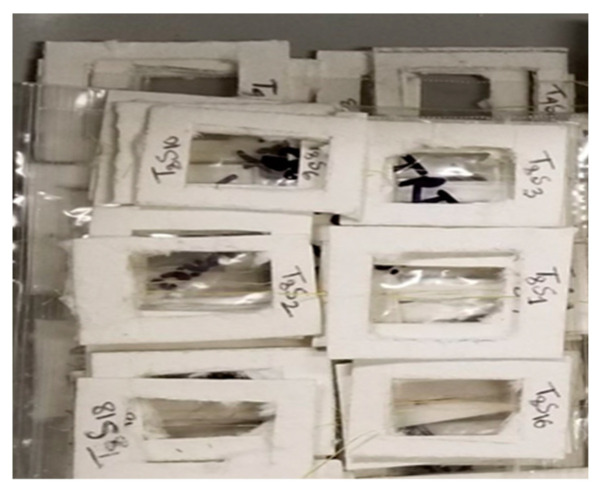
Fibres attached to the frames by glue.

**Figure 2 polymers-14-01685-f002:**
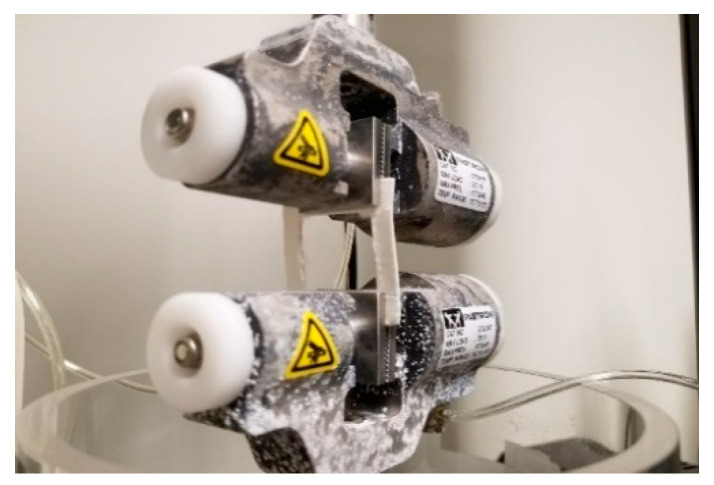
Fibre-frame placed between the jaws.

**Figure 3 polymers-14-01685-f003:**
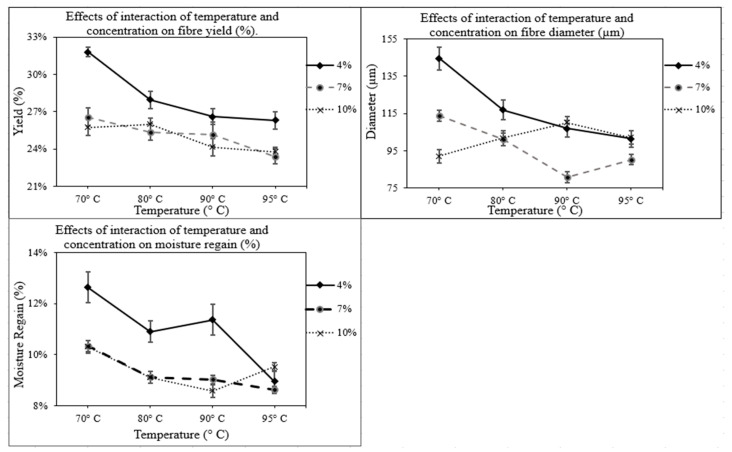
Effects of interaction of temperature and concentration on fibre yield (%), diameter and moisture regain (%).

**Figure 4 polymers-14-01685-f004:**
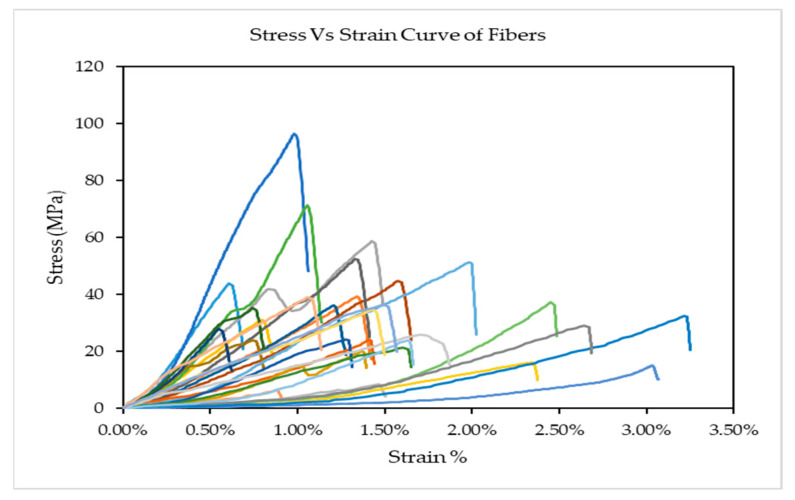
Stress Vs Strain Curve of Cattail fibres.

**Figure 5 polymers-14-01685-f005:**
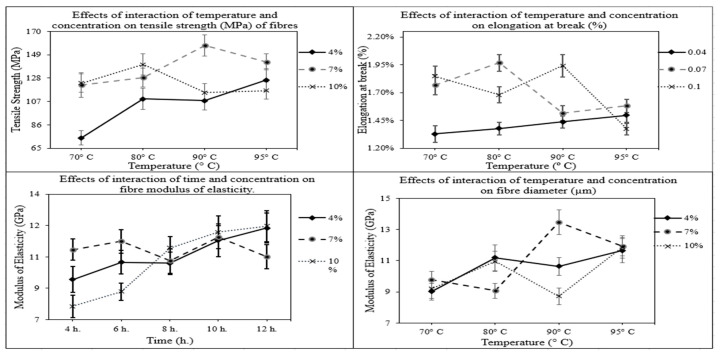
Effects of various interactions on the mechanical properties of fibres.

**Table 1 polymers-14-01685-t001:** Number of replications per treatment, and total number of samples for each response variables.

Response Variables	No. of Replications Per Treatment	Total No. of Treatments	Total No. of Samples
Yield (%)	4	60	240
Diameter (µm)	10		600
Tensile strength (MPa)	10		600
Modulus of elasticity (GPa)	10		600
Elongation at break (%)	10		600
Moisture regain (%)	3		180

**Table 2 polymers-14-01685-t002:** Summary of data and the distribution used for each response variable.

Response Variables	Description of Data	Distribution Used
Yield (%)	Proportions, ranged between 0 to 1 (0 to 100 when expressed as a percentage).	Beta distribution [19]
Diameter (µm)	Highly skewed to the right with skewness = 1.19 and kurtosis = 2.52	Lognormal distribution [19]
Tensile strength (MPa)	Heavily skewed to the right with skewness = 1.43 and kurtosis = 4.66.	Lognormal distribution [19]
Modulus of elasticity (GPa)	Heavily skewed to the right with skewness = 1.57 and kurtosis = 4.18.	Lognormal distribution [19]
Elongation at break (%)	Approximately followed the normal distribution with skewness = 0.66 and Kurtosis = 0.79	Normal distribution [20,21]
Moisture Regain (%)	Approximately followed a normal distribution with skewness = 0.88 and kurtosis = 0.49	Normal distribution [20,21]

**Table 3 polymers-14-01685-t003:** Three-way factorial ANOVA for yield (%) and fibre properties.

	Effects						
Properties	t	H	C	t × H	t × C	H × C	H × t × C
Yield (%)	**	**	**	NS	NS	*	NS
Diameter (µm)	**	**	**	NS	NS	**	NS
Tensile Strength (MPa)	*	*	**	NS	NS	**	NS
MOD (GPa)	**	**	NS	NS	*	**	NS
Elongation at Break (%)	*	*	**	NS	NS	**	NS
Moisture Regain (%)	*	**	**	NS	NS	**	NS

*p* < 0.05 = significant (*); *p* < 0.001 = Highly Significant (**); *p* > 0.05 = Not Significant (NS); MOD: Modulus of elasticity.

**Table 4 polymers-14-01685-t004:** Comparison of cattail fibres with other natural fibres commonly used for composites [6,30,31].

	Properties			
	Strength (MPa)	Modulus of Elasticity (GPa)	Elongation at Break (%)	Moisture Regain (%)
Flax	600–1200	27.6	1.2–3	7
Hemp	690	70	1.6–4.5	8
Sisal	350–370	9.4–19	1.9–3	11
Coir	100–175	6	15–20	10
Cattail (Current Research)	68–169	6.8–15	1.16–2	8–13

**Table 5 polymers-14-01685-t005:** Optimum values of physical and mechanical properties of extracted cattail fibres.

Response Variables	Optimum Values	Response Variables	Optimum Values
Yield (%)	34.57 ± 0.79	MOD (GPa)	14.80 ± 1.26
Diameter (µm)	74.64 ± 2.79	EAB (%)	1.988 ± 0.10
Tensile strength (MPa)	168.5 ± 12.09	MR (%)	7.825 ± 0.16

MOD: Modulus of elasticity; EAB: Elongation at break; MR: Moisture regain.

**Table 6 polymers-14-01685-t006:** The objectives of different properties, their relative weights, target values and tolerance values for automobile applications.

Properties	Objectives	Target Value	Tolerance Value	Weights	Importance Coefficients	Reference Values
Yield (%)	Maximize	34	20	1	1	-
MR (%)	Minimize	8	11	1	1	^a^ 11 (sisal)
Diameter (µm)	Minimize	75	120	1	2	-
TS (MPa)	Maximize	168	100	2	3	^b^ 100–200 (coir)
MOD (GPa)	Maximize	15	10	2	3	^b^ 9.4–19 (sisal)

MR: Moisture regain; TS: Tensile strength; MOD: Modulus of elasticity; ^a^: reference [29], ^b^: reference [36].

**Table 7 polymers-14-01685-t007:** Parameter settings with upper and lower edges for sensitivity analysis.

Parameters	Property (Designation)	Levels	
		Upper (+1)	Lower (−1)
Weight	Yield (%) (A)	2	0.5
	Diameter (B)	2	0.5
	Strength (C)	2	0.5
	Modulus (D)	2	0.5
	Moisture (E)	2	0.5
Importance of coefficient	Yield (%) (F)	5	1
	Diameter (G)	5	1
	Strength (H)	5	1
	Modulus (I)	5	1
	Moisture (J)	5	1
Range	Yield (%) (K)	15–34	25–34
	Diameter (L)	75–140	75–100
	Strength (M)	80–168	120–168
	Modulus (N)	8–14	12–14
	Moisture (O)	8–13	8–9

**Table 8 polymers-14-01685-t008:** The 20 run Plackett–Burman experimental design for 15 factors.

Run	Coded Values for Selected Factors															
	A	B	C	D	E	F	G	H	I	J	K	L	M	N	O	Desirability
1	1	1	1	−1	1	−1	1	−1	−1	−1	−1	1	1	−1	−1	0.21
2	−1	−1	−1	1	1	−1	−1	1	−1	−1	1	1	1	1	−1	0.76
3	1	1	−1	−1	1	−1	−1	1	1	1	1	−1	1	−1	1	0.81
4	1	−1	−1	1	1	1	1	−1	1	−1	1	−1	−1	−1	−1	0.54
5	−1	1	1	1	1	−1	1	−1	1	−1	−1	−1	−1	1	1	0.49
6	1	1	−1	1	−1	1	−1	−1	−1	−1	1	1	−1	−1	1	0.83
7	1	1	1	1	1	1	1	1	1	1	1	1	1	1	1	0.67
8	1	−1	−1	1	−1	−1	1	1	1	1	−1	1	−1	1	−1	0.46
9	1	−1	1	−1	−1	−1	−1	1	1	−1	−1	1	−1	−1	1	0.46
10	−1	−1	1	1	−1	−1	1	−1	−1	1	1	1	1	−1	1	0.93
11	−1	−1	1	−1	−1	1	1	1	1	−1	1	−1	1	−1	−1	0.85
12	−1	−1	−1	−1	1	1	−1	−1	1	−1	−1	1	1	1	1	0.69
13	−1	1	−1	−1	1	1	1	1	−1	1	−1	1	−1	−1	−1	0.27
14	−1	1	1	−1	−1	1	−1	−1	1	1	1	1	−1	1	−1	0.73
15	1	−1	1	−1	1	−1	−1	−1	−1	1	1	−1	−1	1	−1	0.45
16	−1	−1	1	1	1	1	−1	1	−1	1	−1	−1	−1	−1	1	0.45
17	1	−1	−1	−1	−1	1	1	−1	−1	1	−1	−1	1	1	1	0.34
18	−1	1	−1	1	−1	−1	−1	−1	1	1	−1	−1	1	−1	−1	0.23
19	1	1	1	1	−1	1	−1	1	−1	−1	−1	−1	1	1	−1	0.06
20	−1	1	−1	−1	−1	−1	1	1	−1	−1	1	−1	−1	1	1	0.95

**Table 9 polymers-14-01685-t009:** ANOVA for Plackett–Burman experiment.

Source	DF	Sum of Squares	Mean Sum of Squared	F-Ratio	Probability
Model	15	1.214	0.080	6.154	0.0038
Error	4	0.055	0.013		
Total	19	1.269			

## Data Availability

Not applicable.

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
