# Peer review of "Optimization of Typha Fibre Extraction and Properties for Bio-Composite Applications Using Desirability Function Analysis"

_polymers, 2022, doi:10.3390/polym14091685_

Round 1
Reviewer 1 Report
This manuscript presented a study about the extraction of Typha Fibre using several parameters based on alkali extraction. The work need to be deeply improved, as pointed below, since alkali extraction is a common method the remove lignin and hemicellulose from natural fibers.
Section 2.6:how the test was performed? What type of mechanical tests were done? How many specimens were tested? Please be more specific.
Section 3.2 and Section 3.3: I suggest improve the discussion in both sections. What is the effect of the evaluates parameters (time, temperature, NaOH concentration) on the fiber properties? How the combined parameters can significantly affect the fibers properties as can be seen in Table 1?
Tables 4, 5 and 6: please improve the discussion about the results presented in these tables.
Author Response
- Comments: This manuscript presented a study about the extraction of Typha Fibre using several parameters based on alkali extraction. The work needs to be deeply improved, as pointed below, since alkali extraction is a common method the remove lignin and hemicellulose from natural fibers.
Response:
Thanks for taking time in reviewing this document. Appreciate your voluntary time for the good of the society. To improve the work further literature review is added in the Introduction (highlighted in green).
- Comments: Section 2.6: how the test was performed? What type of mechanical tests were done? How many specimens were tested? Please be more specific.
Response:
- A table (Table 1) is added with the experimental details (highlighted in green) in section 2.2.
- Details of mechanical properties are added in the Materials and Methods under section 2.6. (highlighted in green).
- Details about statistical analysis with the equations are added in the Materials and Methods under section 2.8 (highlighted in green).
- Comments: Section 3.2 and Section 3.3: I suggest improving the discussion in both sections. What is the effect of the evaluates parameters (time, temperature, NaOH concentration) on the fiber properties? How can the combined parameters significantly affect the fibers properties as can be seen in Table 1?
Response:
Further discussions have been provided in Sections 3.2 and 3.3 (highlighted in green).
- Comments: Tables 4, 5 and 6: please improve the discussion about the results presented in these tables.
Response:
The discussions of these three tables have been improved and highlighted in green under section 3.4.2.
Reviewer 2 Report
The work "Optimization of Typha Fibre Extraction and Properties For Bio- Composite Applications Using Desirability Function Analysis" is centered on the scope of the journal, interesting, well organized and easy to read. I suggest the publication with some remarks.
In the introduction section I suggest adding some newer references:
A review of wood polymer composites rheology and its implications for processing Mazzanti, V., Mollica, F. Polymers, 2020, 12(10), pp. 1–23, 2304
Lignocellulosic fiber reinforced composites: Progress, performance, properties, applications, and future perspectives Rangappa, S.M., Siengchin, S., Parameswaranpillai, J., Jawaid, M., Ozbakkaloglu, T. 2022 Polymer Composites 43(2), pp. 645-691
Pag 2 line 83-85 The description of the drying verification method needs to be reformulated to increase clarity
Pag 3 line 103 the standard need to be added
Pag 3 line 132-141 I suggest to the authors to add a table with the values indicated in the text to improves clarity
In tensile properties results, a plot stress vs. strain with the different curves need to be added. (at least the most significant)
Pag 8 line 282-285 I suggest to add a table where the link between letter and parameter is specified
Author Response
The work "Optimization of Typha Fibre Extraction and Properties for Bio- Composite Applications Using Desirability Function Analysis" is centered on the scope of the journal, interesting, well organized and easy to read. I suggest the publication with some remarks.
Thanks for taking time in reviewing this document. Appreciate your voluntary time for the good for the society. The response for the specific comment is given below – shown in green colour.
- Comment/suggestion: In the introduction section I suggest adding some newer references:
A review of wood polymer composites rheology and its implications for processing Mazzanti, V., Mollica, F. Polymers, 2020, 12(10), pp. 1–23, 2304
Lignocellulosic fiber reinforced composites: Progress, performance, properties, applications, and future perspectives Rangappa, S.M., Siengchin, S., Parameswaranpillai, J., Jawaid, M., Ozbakkaloglu, T. 2022 Polymer Composites 43(2), pp. 645-691
Response:
The above two references are added in the Introduction section (highlighted in green). The reference numbers are 2 and 3, respectively.
- Comment/suggestion - Pag 2 line 83-85 The description of the drying verification method needs to be reformulated to increase clarity
Response: Added in the Materials and Methods under section 2.3 (Highlighted in Green).
- Comment/suggestion - Pag 3 line 103 the standard need to be added
Response: Standard ASTM D3822 has been added under section 2.6.
- Comment/suggestion - Pag 3 line 132-141 I suggest to the authors to add a table with the values indicated in the text to improves clarity;
Response: A Table (Table 2) is added in Section 3.1 (Highlighted in Green).
- Comment/suggestion - In tensile properties results, a plot stress vs. strain with the different curves need to be added. (at least the most significant)
Response: A figure is added (Figure 4) with the stress-strain curves with explanation under section 3.3. (Highlighted in Green)
- Comment/suggestion - Pag 8 line 282-285 I suggest adding a table where the link between letter and parameter is specified
Response: The relationship has been specified in Table 7 (highlighted in green).
Reviewer 3 Report
This paper leans heavily on statistical analyses of different treatment and responses and is certainly interesting and useful addition. In order to have broader impact, it is recommended that a major revision with an emphasis on the relationship between the measured properties and the final composite properties to be carried out.
- Line 47 “hydroscopic” is an incorrect word. Either hydrophilic or hygroscopic. From the context, the author seemed to mean hydrophilic.
- Line 64 “However, no studies have been done so far to validate the properties of these fibres for their usability in fibre-reinforced composites” There has been a number of studies looking at this directly. Here is a review I found: DOI: 10.1080/15440478.2020.1870643 While the authors claimed to be studying the usability of fiber-reinforced composite, they did not experimentally make or test the composite properties. This gap needs to be filled with additional experiment works or a deeper literature discussion. For bio-composite properties, the bonding between the fiber and the matrix is probably more important than fiber properties itself. However, the authors did not test for or discuss the implication of this important factor.
- The author claimed that the extracted cattail fiber is suitable for automobile applications. Please provide details on what are the polymer matrix commonly used for the automobile applications that the cattail fiber has the potential to be added in as reinforcement? How do the requirements change when other applications such as packaging are intended? Please discuss in details.
- Tensile strength calculation. Is the fiber cross-sectional area calculated from the average fiber diameter of all fibers, the average diameter of the treatment subset, or the average diameter along a single fiber? Have you looked at the dependencies of tensile strength and modulus on fiber diameters?
Round 2
Reviewer 1 Report
After corrections the manuscript quality was improved. I suggest publication in its current form.
Reviewer 3 Report
The authors have addressed my previous comments.